# Home-Field Advantage of Litter Decomposition Faded 8 Years after Spruce Forest Clearcutting in Western Germany

Liyan Zhuang [1], Andrea Schnepf [1], Kirsten Unger [2], Ziyi Liang [1,3,*] and Roland Bol [1]

1   Forschungszentrum Jülich GmbH, IBG-3 (Agrosphäre), 52428 Jülich, Germany; l.zhuang@fz-juelich.de (L.Z.); a.schnepf@fz-juelich.de (A.S.); r.bol@fz-juelich.de (R.B.)
2   Institute of Crop Science and Resource Conservation (INRES), Soil Science and Soil Ecology, University of Bonn, Nussallee 13, 53115 Bonn, Germany; kunger@uni-bonn.de
3   Long-Term Research Station of Alpine Forest Ecosystems, Key Laboratory of Ecological Forestry Engineering, Institute of Ecology and Forestry, Sichuan Agricultural University, Chengdu 611130, China
*   Correspondence: chn_liangzy@163.com or z.liang@fz-juelich.de

**Abstract:** Home-field advantage (HFA) encompasses all the processes leading to faster litter decomposition in the 'home' environment compared to that of 'away' environments. To determine the occurrence of HFA in a forest and adjacent clear-cut, we set up a reciprocal litter decomposition experiment within the forest and clear-cut for two soil types (Cambisols and Gleysols) in temperate Germany. The forest was dominated by Norway spruce (*Picea abies*), whereas forest regeneration of European Beech (*Fagus sylvatica*) after clearcutting was encouraged. Our observation that Norway spruce decomposed faster than European beech in 70-yr-old spruce forest was most likely related to specialized litter-soil interaction under existing spruce, leading to an HFA. Elevated soil moisture and temperature, and promoted litter N release, indicated the rapid change of soil-litter affinity of the original spruce forest even after a short-term regeneration following clearcutting, resulting in faster beech decomposition, particularly in moisture- and nutrient-deficient Cambisols. The divergence between forest and clear-cut in the Cambisol of their litter $\delta^{15}$N values beyond nine months implied litter N decomposition was only initially independent of soil and residual C status. We conclude that clearcutting modifies the litter-field affinity and helps promote the establishment or regeneration of European beech in this and similar forest mountain upland areas.

**Keywords:** clearcutting; Norway spruce; European beech; litter decomposition; N; Ca; home-field advantage (HFA); carbon and nitrogen stable isotopes

## 1. Introduction

Forest cover change contributes to complex feedbacks on forest ecosystems along chronosequences [1] and results in the disruption of ecological processes, including microclimate and soil nutrients mineralization [2–4]. Clearcutting, changing the dominated species, and forest growth dynamics leave behind a significant shift in ecosystem-scale species communities, influencing the decomposition pattern during regeneration. Such as, nutrient-rich litter or logging residual in early successional stages is associated with faster decomposition and turnover rates, while slower organic matter recycling and infertile soil had usually found under older forests [5,6]. The change in the decomposition process determines organic matter sequestration and hence forest growth [7]. A better understanding of litter decomposition and nutrient cycling is necessary for an effective management strategy to promote forest regeneration, especially after deforestation or decades of regeneration [8].

In the last century, large forest areas in central Europe were converted into monocultures of fast-growing spruce. Spruce monocultures are generally known for their low biodiversity and soil deterioration due to acidification and nitrogen leaching [9,10]. To maintain the ecological, sociological, and cultural functions of the forest, the conversion of

existing Norway spruce into more natural broadleaved and mixed forests is the main silvi-cultural aim in Germany and other European countries [11,12]. Some studies have shown that spruce decomposition was accelerated in its originated coniferous stands relative to away from it [13–15]. It is usually considered that soil decomposer organisms may adapt to break down particular substrate in individual ecosystems, thereby accelerating the de-composition of litter from which it is derived (i.e., home) than away from that plant [16,17], which has been termed the home field advantage (HFA) of litter decomposition [18].

Moreover, the data review analysis from Ayres et al. [19] concluded that HFA is widespread in forest ecosystems and suggested that ~30% of the variability of litter decom-position at a global scale can be explained by HFA. Clearcutting brings about a high plant abundance of pioneer species (i.e., high nutrient concentration and low lignin: N ratios) and modified soil abiotic conditions (including nutrient leaching, soil temperature, moisture, and pH) [20–23], resulting in shifts in the functioning of decomposer communities, such as decreased fungal biomass and change in bacterial community structure [24,25]. The resultant association between individual species and site condition can affect soil properties that enhance the decomposition of its own litter, creating an HFA effect for the species-own litter [24]. At the same time, case studies indicated that warmer and moister conditions after clearcutting drive faster litter breakdown by higher soil decomposer activity irrespective of HFA [26,27]. Soil decomposer communities changes when a forest is clear-cut due to the shift in plant communities and soil physical condition, and then microbial differences in ability might arise through local adaption with its "new" home environment (or a 'home' litter) [28], however, studies rarely investigate HFA after removing the dominant species as in clearcuttings. There is a need for gathering reliable scientific knowledge on the influence of clearcutting on original 'home' and 'away' litter decomposition in the new clear-cut.

There is increasing evidence that the strength of HFA is associated with the interaction between local litter quality and specialized microbes. For example, greater fungal biomass in spruce plantations could partly explain the HFA for spruce in its habitat due to the better degradation of recalcitrant fractions through fungi adaption [29]; that is, conifers should favor soil decomposition dominated by fungi and fungivorous microarthropods, in comparison to broadleaved species [30]. Moreover, across succession, soil communities have gone through a wider range of litter qualities contributing to a broader functional capacity to degrade various litter types [28], so decomposer ability in succession may increase with regeneration. However, recent studies pointed out that litter quality was not an important determinant of HFA [31], while the greater dissimilarity between 'home' and 'away' litter indicated strong HFA [32].

Measurements of plant $\delta^{13}C$ and $\delta^{15}N$ abundance have been shown to be useful indi-cators of forest organic matter dynamics [33]. The difference between the isotopic signature of residual litter and litter degradation or litter nutrient dynamics are considered as the inherent tracers for understanding the progression of decomposition and nutrient miner-alization/immobilization [34]. Labile compounds with faster mineralization rates exhibit higher $\delta^{13}C$ values rather than $\delta^{13}C$-depleted recalcitrant lignin [35]. In addition, microbial processes enrich carbon with $\delta^{13}C$ in relation to bulk litter [36]. The changes in foliar$\delta^{15}N$ values are positively associated with nitrate leaching following forest clearcutting [37,38], that is, the foliar $\delta^{15}N$ often relate to N availability, clearcutting increases nitrification and nitrate loss rate, resulting in much of the $\delta^{15}N$-depleted nitrate leaching out, but $\delta^{15}N$-enriched ammonium retaining These findings have provided us with a meaningful point that the alteration of isotopic C and N signature between litter types during decomposition are useful indicators of nutrient status after disruption of the forest.

In the Eifel National Park (Wüstebach, Germany), clearcutting operations were carried out in spruce monoculture in 2013 as the first step of conversion from planted spruce monoculture to natural forest. This significantly affected soil nutrient leaching [39,40], moisture [41] as well as soil respiration [42]. To test the validity of the HFA change long with clearcutting management, we carried out a reciprocal transplant litter decomposition on a 70-yr spruce forest and a clear-cut after short-term (8-yr) regeneration. In addition, we

tested the importance of litter quality on the strength/occurrence of HFA. The difference of litter mass loss and nutrient release, as well as isotopic $\delta^{13}$C and $\delta^{15}$N discrimination between spruce and beech, were determined to figure out this question.

## 2. Materials and Methods

### 2.1. Site Description

The study area is located in Wüstebach (50°30′15.3″ N, 6°20′03.0″ E), situated within the Eifel National Park of western Germany. The climate is mild and humid, with the mean annual air temperature of 7 °C and the mean annual precipitation of about 1200 mm [43]. Winter is moderately cold with periods of snow. Norway spruce replaced European beech as the dominant canopy species for timber production since the 1940s. In the last decades, the Park authority has started accelerating the 'natural' regeneration towards a beech forest by clear-cuts of a significant proportion of the Norway spruce monoculture (~90%) [44]. The ground cover vegetation in these clear-cut stands is formed mainly by young samplings of alder [*Alnus glutinosa* (L.) Gaertn], European beech (*Fagus sylvatica*) with an admixture of early pioneer species, i.e., scrubs, bushes after 8-year regeneration. Norway spruce (*Picea abies*) is the dominant tree species in the remaining uncut forest. Five subplots were selected for this study ranging from 595 m in the northern part to 628 m in the south in forest and clear-cut, respectively. Soils at the stands are classified as Cambisols and Gleysols, and Gleysols nearby stream is moister than Cambisol. For more information about soil properties, refer to Siebers and Kruse. [40] and Wiekenkamp et al. [42].

### 2.2. Litter Decomposition Experiments

Between 2019 and 2020, a reciprocal litter transplant experiment was established in the forest and clear-cut. In September 2019, freshly senesced spruce needles and beech leaves were collected from 6 sampling sites at the forest and clear-cut ecosystems, respectively. Within each collection, each substrate was collected from a minimum of 6 different plant individuals to ensure the representativeness of the pool collected. According to the purpose of forest management, we assumed that spruce is the home environment for the forest, while the home environment for beech is clear-cut.

All samples were air-dried to constant mass. 2.5 g of Spruce needles or Beech leaves were filled into each polyethylene litterbag (10 × 8 cm; 0.25 mm mesh size), respectively. The mesh size permits the entry of bacteria, fungi, and micro-fauna [45]. In October 2019, five sampling locations were selected for clear-cuts and adjacent forests on both Cambisols and Gleysols, respectively. At each subplot, 4 litterbags of each species were placed on the soil surface after getting rid of the humus layer or grass. Litterbags were retrieved after 1, 3, 9, 12 months. Altogether, we prepared 160 litterbags (4 sampling times × 2 stands × 2 soil types × 2 species × 5 replicates) in total. Harvested litterbags were transported to the laboratory. Oven-dried and weighed after removing soil particles and other extraneous materials.

C and N contents of each sampling were measured by a CNS analyzer. The natural abundance of $\delta^{13}$C and $\delta^{15}$N was measured by stable isotope ratios mass spectrometry. The total phosphorus (P), Calcium (Ca) were determined after microwave digestion with $H_2O_2$-$HNO_3$ using inductively coupled plasma mass spectrometry (ICP-MS). The soil temperature and moisture in Wüstebach were measured with the wireless sensor network with 600 ECH2O EC-5 and 300 ECH2O 5TE sensors.

### 2.3. Data Statistics

To determine the strength and direction of home-field effects on litter decay rate, the home-field advantage index (HFAi) for mass loss and the release of C and N was calculated following Ayres et al. [19] and adapted from Veen et al. [31] as

$$\text{HFAi (\%)} = \left( \frac{A_{RL_a} + B_{RL_b}}{2} \Big/ \frac{A_{RL_b} + B_{RL_a}}{2} \right) \times 100 - 100 \qquad (1)$$

where $i_{RL_j}$ represents the relative mass or nutrient loss of species $i$ in environment $j$. Single sample $t$-tests were used to test whether the HFAi differed from 0.

HFAi stands for the additional decomposition or mineralization at home versus away environment and is a net value for both species (A and B) in the reciprocal experiment.

The mean HFA (% increase in $k$ value at home versus away environment) for each litter type was calculated according to [46]:

$$\text{The mean HFA} = (k_{\text{home}} - k_{\text{away}})/k_{\text{away}} \times 100 \tag{2}$$

where $k_{\text{home}}$ and $k_{\text{away}}$ are the decomposition constants of a given species at home and in away environments, respectively.

The $\delta^{13}$C and $\delta^{15}$N values are expressed as

$$\delta\ (\text{‰}) = \left(\frac{R_{sample}}{R_{standard}} - 1\right) \times 1000 \tag{3}$$

where $R_{sample}$ and $R_{standard}$ represent either $^{13}$C: $^{12}$C or $^{15}$N: $^{14}$N ratios of sample and standard material, respectively. The stable isotope ratio values are expressed in parts per million (‰) relative to international standards. Vienna Pee Dee Belemnite (VPDB) for carbon isotope and atmospheric nitrogen for nitrogen isotopes. The analytical precisions for carbon isotopes ±0.1‰ and ±0.3‰ for nitrogen isotopes.

Mass remaining (%) was calculated from dry mass at sampling date divided by initial dry mass. The decomposition rate ($k$ value, yr$^{-1}$) was estimated according to the exponential regression $y = e^{-kt}$, $y$ (%) is mass remaining over time $t$, $k$ is the decomposition rate by Olson. [47]. Nutrients remaining (%) of each sample were estimated as nutrient content at each sampling time divided by initial nutrient content and expressed by % of the initial amount. We performed $t$-tests: (1) to test the significance of initial quality and residuals after one year of decomposition between beech and spruce; (2) to test $k$ values of beech and spruce in clear-cuts and forest on Cambisols and Gleysols; (3) to examine the environmental differences between forest and clear-cut at each sampling point; and finally, (4) to determine if the HFAi was significant between soil types. Repeated measure ANOVAs were used to compare the significance of soil types, stands, and species on various nutrients remaining over time. Three-way ANOVAs were calculated to compare the three factors: soil types, stand, and species on nutrient remaining. A series of stepwise regressions were conducted to detect the variance relationship, like nutrients and stoichiometry on mass loss between soil types, stands, and species. All statistical analysis was performed using SPSS22.0 for the Windows software package.

## 3. Results

### 3.1. Environmental Difference between Stands and Soil

Soil types and forest management greatly influence soil environmental conditions (Figure 1). On average, the soil moisture content was significantly higher in clear-cut than in the forest at both soil types, ranging from 36.6–55%, and Cambisols showed a larger difference in soil moisture by 8.7% than Gleysols by 2.7%, comparing between clear-cut and forest. At both soil types, the temperature at the forest floor was approximately 1 °C higher in clear-cut than in the forest. The results of the $t$-test revealed that the soil moisture and temperature conditions were mostly higher in clear-cut than forest with times, particularly at Cambisols.

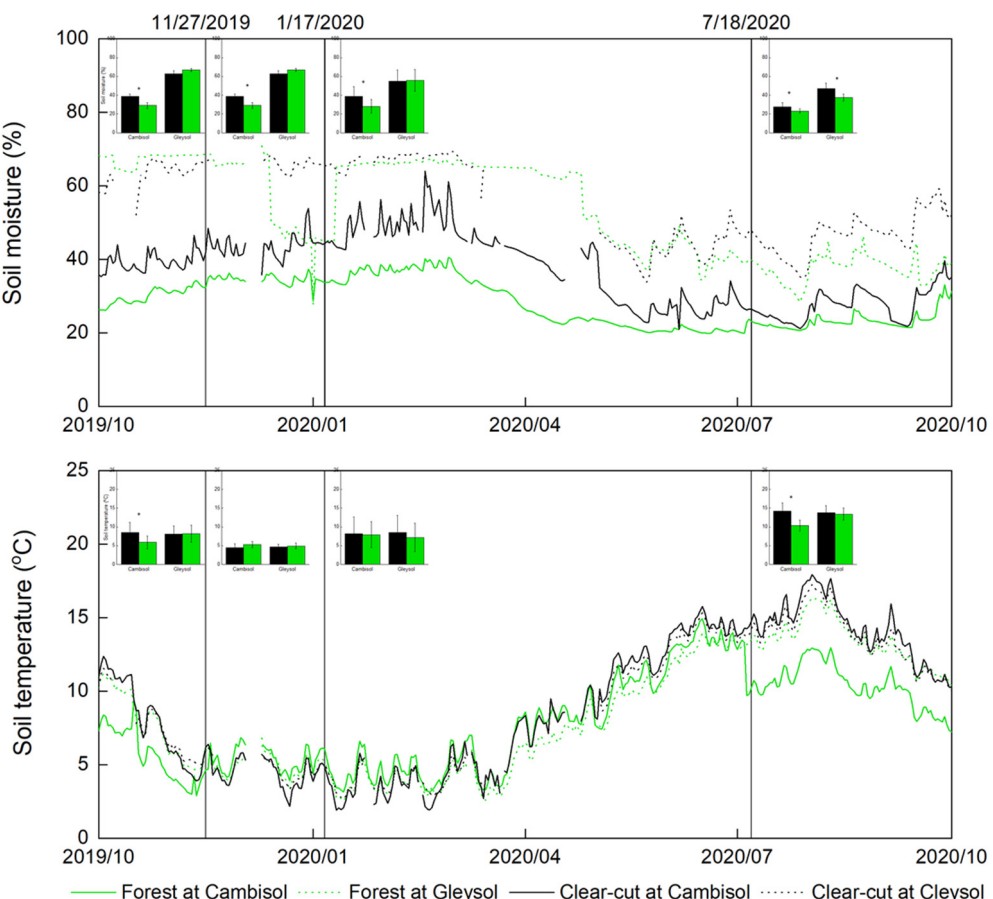

**Figure 1.** The soil moisture (%) and temperature (°C) dynamics in the top layer during one year of decomposition. Bar charts indicate mean values with error bars at each sampling time. The green bar indicated forest; the black bar indicated clear-cut.

### 3.2. Initial Litter Quality and Litter Nutrients after One Year of Decomposition

　　Initial litter quality differed between species. European beech had significantly better initial quality than Norway spruce for C, N, P, and Ca, as well as lower C: N and C:P ratios (Table 1). After one year of decomposition, nutrients concentration and C stoichiometry were significantly different between species and stands. Most nutrient concentrations decreased, except for the N and C:P ratio. Furthermore, C concentration increased in the forest but decreased in clear-cuts for both species during decomposition.

**Table 1.** Nutrient concentrations and compound ratios of beech and spruce litter before and after one year of decomposition.

|  | Initial Litter Quality | | Residual Quality after 1 Year of Decomposition | | | |
|---|---|---|---|---|---|---|
|  | Beech | Spruce | Forest Beech | Spruce | Clear-Cut Beech | Spruce |
| C (%) | 47.1 ± 0.2a | 48.4 ± 0.1b | 48.2 ± 0.8a | 49.5 ± 0.6b | 46.1 ± 0.6a | 47.9 ± 0.5b |
| N (%) | 2.1 ± 0.1a | 1.2 ± 0.0b | 3.0 ± 0.1a | 1.9 ± 0.8b | 2.8 ± 0.1a | 1.6 ± 0.1b |
| P (mg kg$^{-1}$) | 278.4 ± 10.4a | 254.9 ± 8.6b | 105.2 ± 6.3a | 81.0 ± 8.9b | 104.5 ± 6.0a | 61.5 ± 4.9b |
| C:N | 22.4 ± 0.5a | 39.0 ± 1.3b | 20.2 ± 0.6a | 34.4 ± 0.7b | 20.8 ± 0.8a | 35.1 ± 0.4b |
| Ca (mg kg$^{-1}$) | 2.18 ± 0.04a | 1.67 ± 0.01b | 1.02 ± 0.09a | 0.74 ± 0.08b | 1.20 ± 0.05a | 0.79 ± 0.06b |

The lower-case letter indicates the significance between species at the same stands. Different lowercase letters indicate significant differences between beech and spruce in each site ($p < 0.05$).

### 3.3. The Effect of Home-Field Advantage on Litter Decomposition Rates

A significant home-field advantage was shown for the two soil types in this experiment (HFAi = 11 at Cambisols and HFAi = 4 at Gleysols, Table 2). A pattern of the higher decomposition rate of the spruce in the original spruce forest after one year of decomposition followed by *k* values (Figure 2c,d). However, there was no promotion between forest and clear-cut stands in the initial three months. Spruce has a lower 3-month *k* value in forest than in clear-cut (Figure 2a,b), while beech in clear-cut decomposed faster than in forest at Cambisols, but slower when decomposing in Gleysols. Moreover, after one year of decomposition, the *k* value of beech in clear-cut decreased but was higher than the forest stand at Cambisols, while the *k* value of beech in Gleysols did not differ significantly.

**Table 2.** Home-field advantage index of mass loss and C and N release on Cambisol and Gleysol.

|  | Cambisol | Gleysol |
|---|---|---|
| Mass loss | $11.2 \pm 0.5a$ | $3.7 \pm 1.0b$ |
| C release | $14.0 \pm 2.5a$ | $10.7 \pm 0.9b$ |
| N release | $28.3 \pm 0.9a$ | $43.1 \pm 5.4b$ |

The lower-case letter indicates significance between stands. Different lowercase letters indicate significant differences between the two soils ($p < 0.05$).

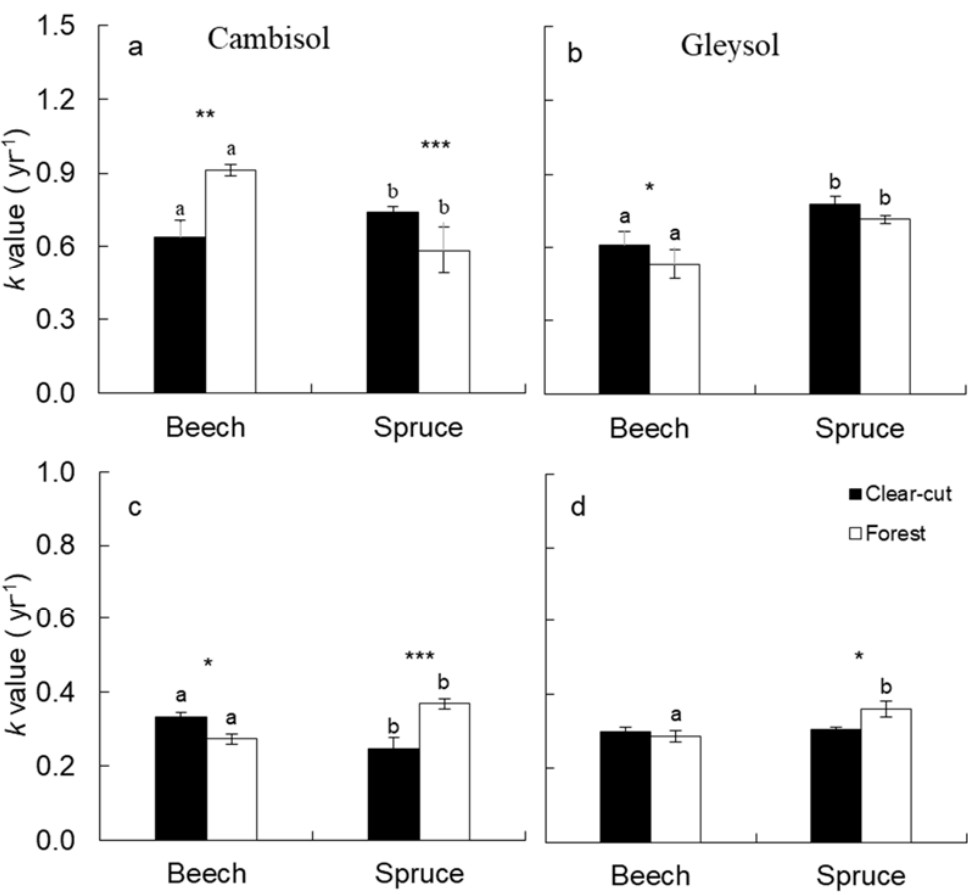

**Figure 2.** k-values (yr-1) of beech and spruce in clear-cut and forest at Cambisols and Gleysols after 3 months (**a,b**) and one year (**c,d**) of decomposition. *: $p < 0.05$; **: $p < 0.01$; ***: $p < 0.001$ indicate significance between stands with same species; lower-case letters indicated significance between species at same stands.

The *k* values for beech and spruce varied among stands and soil types with times. *k* decreased in time for both spruce and beech. The 3-month *k* values were on average 2- to 3-fold higher compared to the 1-year values (Figure 2). The decomposition rate of spruce in the first three months was significantly higher than beech in most stands except for forest stand at Cambisols (Figure 2a,b). Significantly or slightly higher *k* value of spruce showed in forest at all plots except for clear-cut at Cambisols after one year of decomposition, when comparing with beech litter.

### 3.4. C and N Dynamics and Their HFA

Our results indicate that overall C and N release increased at "home" compared with "away" (Table 2). The difference on C release was stronger in Cambisols (14% vs. 10% in Gleysols) for spruce decomposed in forest, while HFAi of N promoted a higher N release in Gleysols (43%) than in Cambisols (28%).

The significance of litter C dynamics varied through time and different treatments (Figure 3a,b, Table 3). A loss of C could be observed in all substrates within the year-long decomposition. Spruce litter lost most C fraction in this original forest during the study period, while beech C in the forest was released rapidly in the first three months and leveled out by the times, which was 6.6% faster on average for spruce in the forest.

**Table 3.** Three-way ANOVA analysis of *F*-value on the effect of soil types, stands, species, and their interactions on nutrient remaining over decomposition.

| Effects | | Remaining | | | |
|---|---|---|---|---|---|
| | df | C | N | P | Ca |
| Soil type | 1 | 6.8 * | 0.7 | 8.4 * | 0.1 |
| Stand × Soil type | 1 | 2.9 | 2.1 | 4.3 | 15.6 ** |
| Soil type × Species | 1 | 31.6 *** | 0.6 | 0.0 | 0.0 |
| Stand × Soil type × Species | 1 | 6.3 * | 1.3 | 8.5 * | 3.2 |

*: $p < 0.05$; **: $p < 0.01$; ***: $p < 0.001$.

Both the spruce and beech did not show significance in the absolute amount of N release in one year period (Figure 3c,d, $p > 0.05$). N immobilization for both species appeared in forest after the first 47 days and almost up to its initial N amount. In clear-cuts, the N remaining generally decreased such that, on average, 9% of the total amount of N was released after one year of decomposition (Figure 3c,d). Irrespective of species and soil types, decomposing litter in clear-cut mineralized relatively more N compared to the initial amount than in forest. There was no significant difference in the retention of N between soil types (Table 3).

### 3.5. The Dynamic of Litter Nutrients Release during Decomposition

Most nutrients indicated significant mineralization over time (Figure 3, Table 3) and observed net mineralization in all substrates following Figure 3. Both leaf litters released P rapidly one year after the start of the decomposition, losing approximately 80% of their initial amount of P (Figure 3e,f, $p < 0.001$). Beech (26%) retained more P than spruce (18%) in one year period ($p < 0.05$). Figure 3e,f shows a similar pattern of P mineralization between forest and clear-cut for both species ($p > 0.05$). On average, the final Ca remaining was significantly higher in forest (70%) than in clear-cut (65%), regardless of species (Figure 3g,h). Furthermore, the interaction between stands and soil types revealed that spruce Ca release was faster than beech (Figure 3g,h, Table 3).

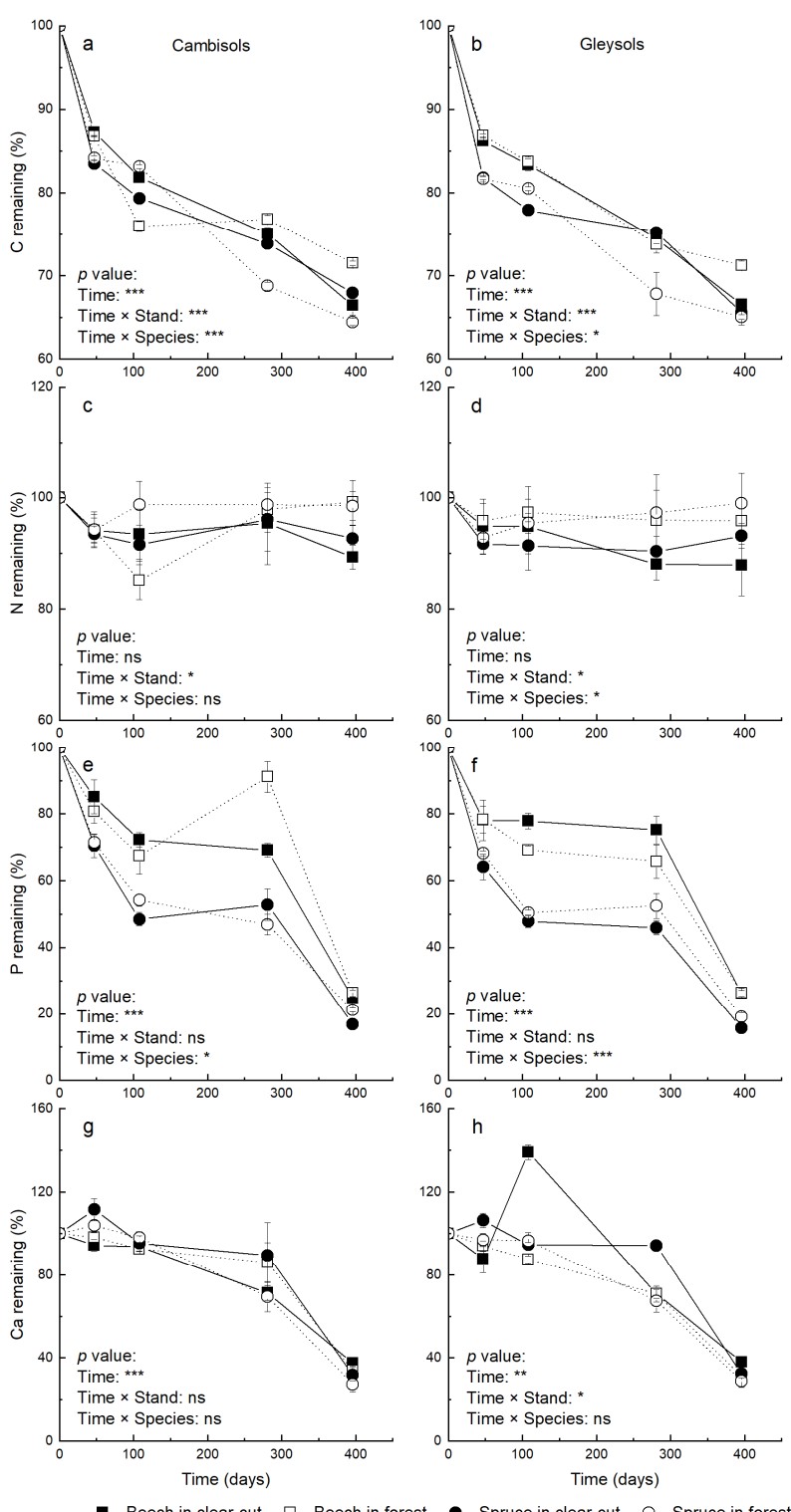

**Figure 3.** Nutrients (C, N, P, and Ca) remaining of beech and spruce in Cambisols (**a,c,e,g**) and Gleysols (**b,d,f,h**) after one year of decomposition. Error bars represent standard errors. Repeated measure ANOVA indicated significance of stand and species with time periods: ns *p* > 0.05, * *p* < 0.05, ** *p* < 0.01,*** *p* < 0.001.

### 3.6. Correlation between Litter Mass Loss Rate and Residual Quality

Litter decomposition rate was associated with changing substrate quality in all sub-plots (Table 4). Decomposition rate of spruce in forest increased with litter N concentration

but decreased with litter Ca concentration ($R^2$ = 0.97). Decomposition of spruce in clear-cut positively changed with N but negatively correlated with P concentration ($R^2$ = 0.93). Beech decomposition rate in forests was positively related to C: N ratios ($R^2$ = 0.67) but also decreased with litter Ca concentration when decomposed in clear-cuts ($R^2$ = 0.85).

**Table 4.** Stepwise regression of the correlation between litter mass loss rate and nutrient concentrations and stoichiometry of beech and spruce under forest and clear-cut over decomposition. Data indicates significant variables related to decomposition, followed by $R^2$.

|  | Variables | Coefficients | $R^2$ |
|---|---|---|---|
| **Spruce** |  |  |  |
| Clear-cut | N, P | 0.65, −0.36 | 0.93 |
| Forest | N, Ca | 0.77, −0.23 | 0.97 |
| **Beech** |  |  |  |
| Clear-cut | Ca, C:N | −0.51, −0.49 | 0.85 |
| Forest | C:N | −0.82 | 0.66 |

### 3.7. Isotopic Change during Decomposition

The $\delta^{13}$C values of decomposing litters leveled off over time across litter types. The initial $\delta^{13}$C values were −32.7 ‰ in beech leaves and increased by 0.13 ‰ on average. For spruce needles, the initial $\delta^{13}$C value was −30.6 ‰ and decreased by 0.14 after 1 year of decomposition. The initial $\delta^{15}$N values ranged from −3.3 ‰ in beech leaves and −4.7‰ in spruce needles (Figure 4). The $\delta^{15}$N values for both species were finally higher in clear-cut than in forest. In the first 9 months, $\delta^{15}$N became enriched in all subplots but then depleted in forest, while a larger decrease happened in Cambisols. Over the same period, the $\delta^{15}$N value in clear-cut became higher throughout the experimental period in Cambisols (−3.0 and −3.7 ‰ for beech and spruce, respectively), but it slightly dropped since July in moister Gleysols (−3.2 and −4.1 ‰ for beech and spruce, respectively) (Figure 4). Linear regression plots of N isotopic against C concentration (%) were negatively significant among species over both stands ($p < 0.05$, Figure S2). While the relationship of the $\delta^{13}$C values and C: N was only linearly significant in forest for spruce ($R^2$ = 0.83, $p < 0.01$).

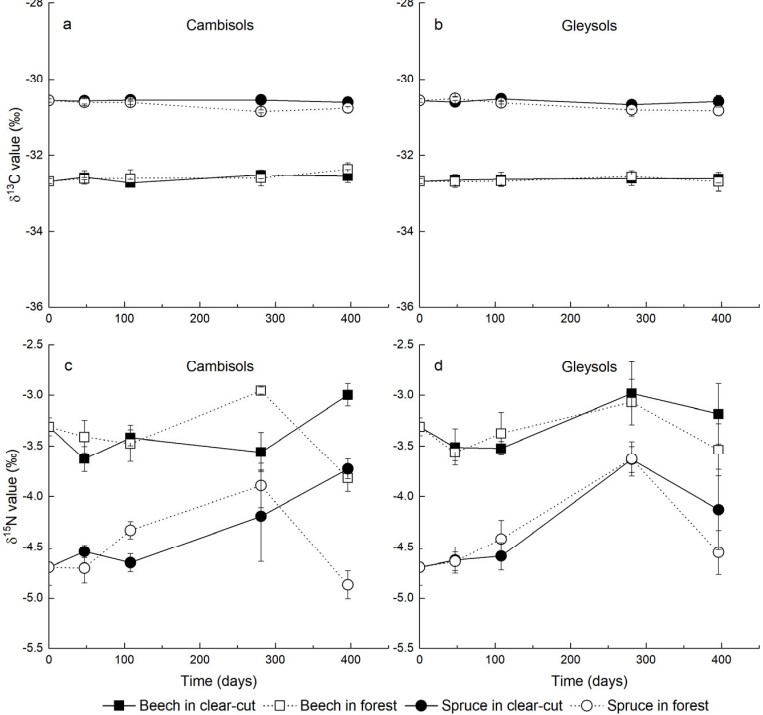

**Figure 4.** The change of isotopic $\delta^{13}$C and $\delta^{15}$N value of beech and spruce over one year of decomposition. Error bars represent standard errors.

## 4. Discussion

### 4.1. HFA in Forest and Post-Harvest Decomposition

Our results indicated a positive effect of litter-site interaction on litter decomposition rate at the home of 11% and 4% and thus a net HFA in the spruce forest, which verified our first hypothesis. A meta-analysis has noticed that decomposition HFA is widespread in forest ecosystems, with on average 4.2% promotion in the home habitat [46]. Low-quality spruce with low nutrient content and high C:N decomposed faster in spruce forest, probably due to the presence of more fungal communities well adapted for degrading recalcitrant litter [31]. After 8-year clearcutting, both soil physical (soil moisture and temperature, Figure 1) and chemical conditions [40] had been markedly elevated, which can influence the development and succession of microorganisms that can assimilate substrate [48]. And thus, we found a significant suppression on spruce decomposition and a slight promotion on beech decomposition in clear-cut. On the other hand, post-harvest regeneration of understory species improves soil nutrient availability and forest sites quality, masking the original soil-litter affinity on pre-harvest forest [49]. This could also account for the lower decomposition rate for spruce in the clear-cut and potentially masked the mean HFA for spruce after a short-term regeneration.

### 4.2. Litter Chemistry Regulated Decomposition of Norway Spruce in Original Forest

Decomposition and mineralization in the initial phase are generally characterized by the leaching of soluble nutrients and by decomposition of soluble and non-lignified cellulose and hemicellulose [50]. Winter snow cover and snowmelt in this initial period physically breakdown litter tissue and accelerated nutrients release and mineralization [51], resulting in a higher *k* value in clear-cut versus forest, and thus no HFA was detected in the initial 3-month decomposition.

The decomposition difference between litter types was correlated to the concentration of C and N, and C: N ratios. Our results also corroborated this hypothesis that litter N concentration served as the most critical nutrient to regulate the degradation of spruce, and beech was decreased with increased C: N ratios, according to the stepwise regression (Table 4). Slower N release was detected in forest, which decreased litter C:N ratios and promoted the generation of brown and white rot fungi [52], and benefited the degradation of the lignin-rich substrate (i.e., spruce). Faster spruce Ca release strengthen the soil acidification that maintains the soil pH, sustaining the home-field effect. Although litter quality well-regulated the litter mass loss, litter quality independently did not serve as a predictor of mean HFA in this case (supplementary, Figure S1). This result is supported by evidence from Veen et al. [31]. This may be because HFA is not restricted by single litter types, but the heterogeneity of litter quality between the 'home' and 'away' habitats [32]. Alternatively, the occurrence of HFA is likely system-dependent, suggesting that transplants between labile litter from nutrient-rich ecosystems and recalcitrant litter from nutrient-limited ecosystems better induce HFA [53,54]. The results from this work were limited to spruce and beech only; a wider assessment between species and ecosystems is necessary for relevant controlling to determine the magnitude and the direction of HFA for plant traits.

### 4.3. Clearcutting Promoted Beech Decomposition and Nutrient Release Pattern

Beech leaves decomposed faster during the first year in clear-cut, which is accompanied by an increase in the mineralization rate of C and N in beech leaves and higher in immobilization in spruce needles. In addition, the less home effect of C, N release was observed after clearcutting. Thus, a transfer from spruce to beech would facilitate the potential utilization of nutrients by trees. The shift of dominated trees species by clearcutting treatment would inherently influence the regeneration in this site through litter input quality [40].

Decay rates for beech in both stands were tightly related to C: N ratios. Beech with lower C: N ratios contributed to a faster decomposition rate for beech (*k* = 0.31 on average) than spruce (*k* = 0.29 on average) in clear-cuts. Changing environmental conditions would

directly affect litter mass loss after rapid shifts in plant community composition [32], contributing to the relatively elevated mass-loss rate for beech in clear-cuts than in spruce forests and the suppression of mean HFA effect for spruce. Moreover, removing the forest canopy elevated atmosphere C and N deposition with precipitation promotes soil nutrient availability in the short term [40] and restructures the local fungal community in soil [25]. This would further hinder the litter decomposition and nutrient turnover rate in these successional stands.

*4.4. Soil Moisture as a Mediator of Litter Decomposition and HFA*

Our results indicated that the decay of beech in clear-cuts differed between soil conditions; that is, beneath Cambisols, the decay rates of beech were significantly higher in clear-cuts than in the forest, and interestingly, it was faster than spruce in clear-cuts. However, a minor difference in clear-cut was observed when decomposing on Gleysols, as well as a decline in the HFA of decomposition and C concentration. Gleysols nearby the stream is moister than Cambisols. The microbial breakdown is likely limited with a high soil moisture level [55], probably resulting in insignificant decomposition between species and stands. Additionally, given the importance of the water-driven decomposition determines a weak mass loss in low-quality litter [56,57], contributing to a similar *k* value (from spruce) between soil types.

Across soil types, the results showed lower HFA on mass and C on Gleysol. A saturated soil environment has been identified to reduce soil microbial decomposition [58]. Soil microbial communities of high soil moisture are generally N limited due to the less nutrient availability [59], resulting in higher N accumulation and N release HFA in Gleysol.

*4.5. Dynamics of the Natural Abundance of $\delta^{13}$C and $\delta^{15}$N during Decomposition*

Isotopic discrimination during litter decomposition has been observed in several studies involving selective consumption of various C compounds. Litter C concentration with $\delta^{15}$N value, in this case, was negatively significant. This correlation signifies that $\delta^{15}$N discrimination between litter types is due to the preferential recalcitrant fraction in substrates, which is consistent with several studies [35,60]. Microbial analysis suggests that $^{15}$N was transferred actively aboveground by saprotrophic fungi [61] via promotion in the lignin or tannin degradation by fungi-based microbes. Suggesting that decreased $^{15}$N values by retaining more litter N from forest floor than from clear-cut do contribute to higher microbial uptake and hence faster spruce litter C degradation in 'home' forest, strengthening the HFA.

In our study, we found a negligible change of $\delta^{13}$C between stands during decomposition; residual C pools with slightly $\delta^{13}$C distinct were needed to account for the duration of the experiment. A report from Ngao and Cotrufo. [62] indicated litter $\delta^{13}$C discrimination appeared particularly in late stages of litter decomposition owing to the increase in the $\delta^{13}$C of decomposition litter $\alpha$-cellulose. Future long-term litter decomposition studies on the discrimination of natural abundance of isotope between species types and ecosystems are therefore recommended.

**5. Conclusions**

Spruce decomposed faster in spruce forest while beech decomposed faster in clear-cut, tightly associating with litter quality, indicating the occurrence of decomposition HFA at forest and clear-cut. Promoted decomposition and C mineralization for spruce in forest could be implied through relatively higher residual N concentration. Since the clear-cut in 2013, plant community and soil environment had shifted the historical resources from the original forest that facilitated faster beech decomposition and nutrients turnover rates due to lower C:N, thereby overriding pre-existing species HFA effects, especially at dryer Cambisols. $\delta^{15}$N diverged after nine months at Cambisol between forest and clear-cut, suggesting that litter N decomposition correlated to soil and residual C status. This has implications for the management of upland forests that are currently still under conifers:



Their regeneration to more natural forests with European beech can be promoted in short-term by intensive forest management.

**Supplementary Materials:** The following supporting information can be downloaded at: https://www.mdpi.com/article/10.3390/soilsystems6010026/s1, Figure S1: The relationship between total C, N concentrations, δ13C, δ15N, C:N ratios, Ca and mean HFA; Figure S2: The relationship between the initial litter C, N and C:N ratios on the isotopic 13C and δ15N value.

**Author Contributions:** Designed study concept: L.Z., R.B. and A.S. Sample preparation, field sampling, and obtained data: L.Z., Z.L. and K.U. Wrote and revised the text: L.Z., R.B. and A.S. All authors have read and agreed to the published version of the manuscript.

**Funding:** Liyan Zhuang and Ziyi Liang acknowledge Ph.D grant support from China Scholarship Council, overall project support was also via German Research Foundation (DFG) under Germany's Excellence Strategy, EXC-2070—390732324—"PhenoRob" and Helmholtz Association grant 2173 Agro-biogeosystems: controls, feedbacks and impact (POF IV: 2021-2026).

**Institutional Review Board Statement:** Not applicable.

**Informed Consent Statement:** Not applicable.

**Data Availability Statement:** The data are available from the authors upon request.

**Conflicts of Interest:** The authors declare no conflict of interest.

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
