# Peer review of "Home-Field Advantage of Litter Decomposition Faded 8 Years after Spruce Forest Clearcutting in Western Germany"

_soilsystems, doi:10.3390/soilsystems6010026_

Round 1

Reviewer 1 Report

By conducting a reciprocal transplant litter decomposition experiment in 70-yr-old planted spruce forest and 8-yr-succession clear-cut, which both contained two soil types, the authors studied the home field advantage (HFA) effect on the litter decomposition of spruce and beech (natural succession species), and nutrients dynamics during the decomposition. The results provide consequential implications for the forest management of central Europe, which suggested regeneration and succession to more healthier forests can be promoted by forest management. The experiment & data analysis is reasonable and results are reliable. I do not have major methodological problem with this manuscript, except the methods need clarification. However, for the question on HFA, the contents in the introduction and discussion do not meet the contents of the title and abstract. Meanwhile, the results are scattered and there are abundant flaws in the presentation. As litter decomposition is depended on the litter-field interactions (which could generate HFA) and the relationships also change with environment, a more structured presentation is needed to interpret the results and the corresponding analysis & discussion on this main scientific question.

Major points

Be careful using the “HFA” throughout the article, in many scenes in the paper talking about the spruce litter decomposition, it is just the decomposition rate in some place. HFA is a simple definition of phenomena, it denote the difference of decomposition at “home” and “away”, the authoritatively accepted mechanism behind is the litter-field affinity. When quantitatively discuss the HFA according to your results, it should be referred to the HFAi or the mean HFA (or recommended ADHi), for the latter, the species should be indicated, e.g., mean HFA of the spruce.

Title

“HFA faded 5-10 years after clearcutting”, it looks more like a conclusion describing a time series study, e.g., spruce litter decomposed in different forest stands with a serious of secondary successions after clear cutting. However, the study in fact only has one time (8 yr after clear-cut). More specific place should also be accounted.

Abstract

Line 20: There is no direct data support on the “microbial biome” in this study.

Line 21-22: “Faster beech decomposition mask HFA for spruce”, why & how? Here is just the disadvantage of away incubation of spruce. If there was anything that could be masked, it would be referred to the changed litter-soil affinity, i.e., clearcutting lead to soil decomposer community succession in favour of decomposing beech.

Line 25: “modifies HFA” may also be replaced with “modifies the litter-field relationship” or “modifies the litter-field affinity”.

Introduction

Line 35: “influence” → influences

Line 39: delete “in forest ecosystems”

Line 50: from “it”? faulty wording or formulation

Line 57: “As such” means what? Delete it would be better

Line 61: “higher”, do you meant larger or more abundance? or higher biodiversity? The literature cited here just proof a restructuring of microbial community after clearcutting

Line 63: delete “with plant-soil interactions”

Line 67: “develops with regeneration through local adaption or dispersal”, how? the mechanisms illustrated is ambiguous and vague.

Line 69: “there” → There

Line 73-76: The viewpoint seemed not precise enough. For the litter decomposition, whether fungi or bacteria were more dominant is more likely depended on the decomposition stage, i.e., the microbial community shift through time. And the litter cited seemed nether support enough.

Line 75: “ro” → to

Line 78-79: “HFA may increase with regeneration”, here, what "may increase" should be the decomposer ability in the succession, but not the HFA (according to its definition).

Line 86-87: “Labile compounds” turn over “slower”?

Line 90-93: unclear expression, reframe the representation.

Line 100-101: “.” → ,

Line 101: “field litter decomposition” → reciprocal transplant litter decomposition

Materials and methods

Line 133-134: 10 incubation stands in total? and 2 soil type in each stand? should be clear

Line 137: If you meant there are two soil types in each of the two forest type, the total factor experiment should be clear. 

Line 155: I recommend ADH here, i.e., the percentage of additional decomposition at home (Ayres et al. (2009) and Giesselmann et al. (2011))

Line 160: “anf” → and

Line 177: “Stepwise regression analysis indicated” better changes to “A serious of stepwise regressions were conducted to detect”

Line 184: the second “forest” → clear-cut

Results

Fig. 3. In this study, statistical test on the differences between “home” and “away” may make more sense than just on the forest stand, also their interaction with other factor such as time and species.

Line 277: “The initial δ13C values were -32.7 ‰”, the “initial” value should be “changed” to a specific figure.

Discussion

Line 301: be careful on “dominated by bacterial communities”. Generally, beech litter is also a hard species, fungi were commonly played more functional roles during the decomposition of litter cellulose and lignin.

Line 307: “lack of mean HFA for spruce in the clear-cut”, how can a HFA be discussed of by one species away, the presentation should be such as: weak HFA, or lower decomposition rate in someplace.

Line 307-310: restate the last sentence, to long to be logical.

Line 333-334: it seemed the words didn't finished.

Line 347: “HFA effect for spruce”. As the main point throughout the paper, if the question here is the ADH of spruce, it would not be suppressed by another species, be careful of the difference between HFA and decomposition rate of one species.

Line 362: delete ” the degradation of”.

Reviewer 2 Report

Comments on the manuscript  entitled „Home-field advantage of litter decomposition faded 5-10 years 2 after spruce forest clearcutting”

Manuscript number: soilsystems-1601906

General comments

In general, the manuscript concerns on the impact of forest management in the Eifel National Park in Germany (spruce monoculture and clearcutting operations) on the biodegradation of the lignocellulosic complex present in spruce needles and beech leaves (litter) measured by changes of various chemical and physical parameters. These studies were aimed not only at assessing the changes taking place in the litter decomposed by microorganisms at forest and in clear-cut ecosystems but also the designation of the occurrence of HFA (Home-field advantage of liter) in two soil types (Cambisols and Gleysols).

Specific comments

Abstract

  1. Sentence starting with “ Our observation…” should be changed because authors not determined biodegradation of spruce or beech (as wood) only needles of spruce and beech leaves in two different soils and different systems.
  2. Authors should changed words“microbial biom” onto “ microbiome”
  3. Page 3, line 109: In which country are there the Eifel National Park Country? This information should be mentioned in the section “Materials and methods”.
  4. Line 135-136: please specify in centimeters what layer it was
  5. Line 189, 191: should be Cambisols
  6. Line 143: How was the humidity and temperature measured. The authors describe these parameters in the Results chapter.
  7. Line 312: please specify which component of litter are biodegraded? np. decomposition and mineralization of spruce needles or beech leaves?
  8. Line 323: It is „brown rot and white rot”, should be “brown and white rot fungi”

“fasten” or ‘faster”

Round 2

Reviewer 1 Report

The manuscript has been carefully revised, and I feel that the current version is ready to be published in Soil Systems.